# Multisystemic Manifestations in Rare Diseases: The Experience of Dyskeratosis Congenita

**DOI:** 10.3390/genes13030496

**Published:** 2022-03-11

**Authors:** Michele Callea, Diego Martinelli, Francisco Cammarata-Scalisi, Chiara Grimaldi, Houweyda Jilani, Piercesare Grimaldi, Colin Eric Willoughby, Antonino Morabito

**Affiliations:** 1Pediatric Dentistry and Special Dental Care Unit, Meyer Children’s University Hospital, 50139 Florence, Italy; 2Unit of Metabolism, Bambino Gesù Children’s Research Hospital, Piazza Sant’Onofrio, 4, 00165 Rome, Italy; diego.martinelli@opbg.net; 3Servicio de Pediatría, Hospital Regional de Antofagasta, Antofagasta 1240835, Chile; francocammarata19@gmail.com; 4Department of Pediatric Surgery, Meyer Children’s Hospital, Viale Gaetano Pieraccini 24, 50139 Florence, Italy; chiara.grimaldi@meyer.it; 5Genetic Department, Mongi Slim Hospital, Marsa 2046, Tunisia; houweyda.jilani@rns.tn; 6Faculty of Medicine of Tunis, University of Tunis El Manar, Tunis 1068, Tunisia; 7Department of Public Health and Pediatric Sciences, University of Torino, 10125 Torino, Italy; pcgrimaldi@gmail.com; 8Genomic Medicine, Biomedical Sciences Research Institute, Ulster University, Coleraine Campus, Coleraine BT52 1SA, UK; c.willoughby@ulster.ac.uk; 9Department of Neurofarba, University of Florence, Viale Pieraccini 6, 50121 Florence, Italy

**Keywords:** dyskeratosis congenita, telomeropathies, clinic, etiology, treatment

## Abstract

Dyskeratosis congenital (DC) is the first genetic syndrome described among telomeropathies. Its classical phenotype is characterized by the mucocutaneous triad of reticulated pigmentation of skin lace, nail dystrophy and oral leukoplakia. The clinical presentation, however, is heterogeneous and serious clinical complications include bone marrow failure, hematological and solid tumors. It may also involve immunodeficiencies, dental, pulmonary and liver disorders, and other minor complication. Dyskeratosis congenita shows marked genetic heterogeneity, as at least 14 genes are responsible for the shortening of telomeres characteristic of this disease. This review discusses clinical characteristics, molecular genetics, disease evolution, available therapeutic options and differential diagnosis of dyskeratosis congenita to provide an interdisciplinary and personalized medical assessment that includes family genetic counseling.

## 1. Clinical Aspects

Disorders in the biology of telomeres or telomeropathies comprise a number of genetic defects, of which dyskeratosis congenita (DC) was the first reported entity [1]. DC presents with a mucocutaneous triad of skin reticulated lace pigmentation [2], principally involving the neck area and the upper anterior thorax, nail dystrophy and oral leukoplakia. The clinical phenotype has expanded considerably since its initial description.

Initial dermatological signs appear in the first years; the clinical picture, however, can be heterogeneous [3]. Nail dystrophy involves at the beginning of the fingernails, then it starts with grooves and longitudinal divisions, evolving in rudimentary, small or absent nails. Leukoplakia impacts the oral mucosa, the tongue and the oropharynx [4]. Palmoplantar hyperkeratosis can lead to painful fissures and ulcers [5,6]. Approximately 30% of patients present malignant transformation to squamous cell carcinoma, thus necessitating carcinogenic surveillance, even with the execution of frequent biopsies in the involved areas [4].

Later on, it was described that DC could affect several organs prematurely, such as bone marrow failure [7], representing the most serious clinical complication. Bone marrow failure can be present in up to 80–90% of subjects when they reach their thirties and may cause more than 70% of deaths in DC patients [8]. The typical mucocutaneous triad guides the differential diagnosis between DC and other causes of bone marrow failure. DC also results in immunodeficiencies, predisposition to dental caries, hypodontia, recession and bone loss, requiring it to be differentiated from juvenile periodontitis, taurodontism, gingival inflammation, and brown intraoral pigmentation [9,10]. Presence of permanent teeth with a decreased root/crown ratio may indicate a diagnosis of DC [11]. Other disease manifestations are pulmonary fibrosis [12] and liver failure or fibrosis [13]. The risk of developing hematological and solid tumors is 50 times higher in patients with DC compared to general population; these include myelodysplastic syndrome, acute myeloid leukemia, non-Hodgkins lymphoma, cell carcinoma squamous cells of the head and neck, esophagus, anogenital cancer and basal cell carcinoma [14,15]. The most common solid tumor in DC is head and neck squamous cell carcinoma, which may present at a younger age compared to the general population (mean age of onset 32 years compared to 67 years); 50% of the patients who developed solid tumors were shown to carry a mutation in the *TERC* gene (Table 1).

DC can also present other “minor” features, i.e., intrauterine growth retardation [16], psychomotor delay, microcephaly [17], premature aging, early graying of hair [3] and short stature [18]. Some patients with DC also present with neurocognitive and neuropsychiatric manifestations, like adjustment disorders, anxiety disorders, ADHD, intellectual disability, mood disorders and schizophrenia [19]. Shorter telomeres seem to be associated with certain psychiatric disorders [20,21,22], but a direct biological link is still lacking. It is still not clear whether short telomeres predispose patients to develop certain neuropsychiatric conditions, and this hypothesis requires further studies.

Ocular anomalies include nasolacrimal duct stenosis, epiphora, blepharitis, sparse eyelashes, ectropion, entropion and trichiasis [23]. Retinal changes are rarely described, mainly hemorrhages, infarction of the nerve fiber layer, macular edema, preretinal fibrosis and optic atrophy. In addition, cardiomyopathy, malabsorption enteropathy, esophageal and urethral stenosis [24,25], hypothyroidism, hypogonadism, testicular atrophy [26], osteoporosis [27] and avascular necrosis of the shoulder and hip joints [28,29] may also be detected. These clinical manifestations affect patients in different ways in a variable number of cases. DC is often diagnosed late due to of the fact that mucocutaneous findings are not present in all cases. The wide spectrum of clinical presentation and the lack of conclusive laboratory tests can sometimes make clinical diagnosis challenging [30].

As in other telomere diseases, the severity of symptoms in DC correlates with the degree of telomere shortening. Most severe cases (i.e., those with greater telomere shortening 1% of the population), DC manifests in the first 10 years of life or even during pregnancy. In the less severe cases of telomeric shortening (<10% percentile), the age of onset is between 15 and 25 years; in these patients, DC can present as medullary aplasia or pulmonary fibrosis, with different probability according to the mutated gene [31]. Considering the low efficacy of therapeutic options, it is crucial to reach an early genetic diagnosis; the common genetic anticipation in DC requires a timely diagnosis also for family genetic counseling [30].

## 2. Related Disorders

The Hoyeraal-Hreidarsson syndrome (OMIM #305000) is the most severe form of DC, showing progressive marrow failure, intrauterine growth retardation, developmental delay, microcephaly, cerebellar hypoplasia, delayed myelination, hydrocephalus, brain atrophy, calcification [32], mental retardation, progressive immunodeficiency and the mucocutaneous triad [33]. The diagnosis is made if the subject presents four or more of by the association of cerebellar hypoplasia with additional signs of DC. Mucocutaneous symptoms and cerebral calcifications may be present [34]. Hoyeraal-Hreidarsson syndrome usually leads to death in early childhood and results from mutations in the *RTEL1* and *DKC1* genes, causing a decrease in telomerase activity [35,36,37]. It can also be due to pathogenic variants in the *TERT*, *TINF2, TPP1 PARN* genes [38,39,40,41].

Revesz syndrome (OMIM #268130) was described for the first time in 1992 [42]. It is another infrequent variant of DC due to pathogenic variants in the *TINF2* gene [39]. The representing symptoms are the presence of bilateral exudative retinopathy, which is associated in most cases with intracranial calcification, and the classic alterations of DC, such as early bone marrow failure [43] and mucocutaneous disease. Intrauterine growth retardation, cerebellar hypoplasia and developmental delay may also be present [42].

Coats plus syndrome (OMIM #612199) is a rare AR disease due to pathogenic variants in the *CTC1* gene [35], with cerebroretinal microangiopathy, intracranial calcifications, brain cysts, leukodystrophy, osteopenia, bone fractures and poor bone healing [44] and gastrointestinal bleeding [45,46].

## 3. Genotype–Phenotype Correlations

The genotype–phenotype correlation is made highly difficult by several factors, such as the possibility of hypomorphic gene mutations, disease anticipation and genetic and environmental modifying factors [13].

Heterozygous AD pathogenic variants in the *TERT* gene may be associated with adult-onset isolated bone marrow failure or pulmonary fibrosis. Compound heterozygous mutations in *WRAP53* present the classic DC with the mucocutaneous triad plus or minus tongue squamous cell cancer [47,48,49].

The clinical symptoms associated with *CTC1* pathogenic variants may not show the mucocutaneous symptoms; on the contrary, cytopenias, retinal exudates, intracranial calcifications or cysts, ataxia, intrauterine growth retardation, osteopenia and/or poor bone healing are frequent

DC patients without the pathogenic variant in one of the eleven known genes may present the most clinically severe phenotypes, including multiple clinical symptoms of DC, Hoyeraal-Hreisdarsson syndrome or Revesz syndrome [31,36,37,40].

## 4. Molecular Bases and Diagnosis

Telomeres are made up of six nucleotide repeats at the ends of chromosomes and a group of nucleoproteins located in these sequences, essential for chromosomal stability [50]. Each time a cell divides, they shorten [51]. The control of telomere length is involved in cell aging, tumorigenesis, so germline mutations in genes involved in telomere maintenance machinery may cause clinical entities like DC [52].

After the discoveries of these disorders, diagnostics test such as flow cytometry and fluorescent in situ hybridization in leukocytes have been developed [47,48,49]. A reduction in the size of telomeres in leukocytes under the first percentile for age is more than 95% sensitive and specific for DC patients [31] compared to unaffected family members or patients with other genetic causes of marrow failure. Other than being involved in the diagnosis of DC, telomere length has directed the discovery of genes associated with the development of DC [47,48,49].

Other methods employed in the diagnosis of DC s to measure the size of telomeres are quantitative polymerase chain reaction and Southern blot. Nevertheless, careful consideration is requested when selecting the method for measurement of telomeres size in a research and clinical setting [53].

A combination of candidate gene sequencing, linkage studies and, more recently, whole exome sequencing have identified at the moment 14 genes involved in telomere shortening involved with DC or similar phenotypes (Table 1) [54]. These genetic defects represent between 70–80% of patients with DC [55]. However, the genetic basis is unknown in 20–40% of cases [56].

DC exhibits diverse inheritance pattern including X-linked recessive (OMIM #305000) [16], as well as autosomal dominant (OMIM #127550) [57] and/or recessive (OMIM #224230) [58] inheritance patterns. Recessive X-linked DC (X-DC) is caused by pathogenic variants in the dyskerin 1 gene (*DKC1*), located in Xq28, which encodes a pseudouridine synthase [1]. X-DC occurs in males with an onset age between 5 and 12 years. Nevertheless, a variety of ages at onset, symptoms and severity may occur, even in subjects with the same mutation, making diagnosis highly complex. In X-DC, women may exhibit less severe clinical features due to lyonization, but always present at an older age [59].

The *DKC1* gene has a highly conserved sequence required for the binding to small nucleolar RNAs, involved in ribosome biogenesis, and participates to telomerase complex at the moment of binding telomerase RNA or *TERC* [57]. In 2002, Mitchell and Collins first connected telomeres and human disease showing aberrant function of dyskerin and shortening of telomeres [60]. Therefore, germline mutations in genes involved in telomere maintenance result in abnormal shortening of these structures compared to age-matched controls, leading to chromosomal instability and progressive cell death [56].

As a result, DC represents a disease of defective telomere maintenance, leading to premature shortening, replicative senescence, premature depletion of stem cells and multisystem involvement. DC involves more prominently highly proliferating mucocutaneous tissues [61]. However, defects in telomerase and telomere components have also been shown in subjects with aplastic anemia, pulmonary fibrosis and liver disease [62].

Despite the strong link between DC and shorter telomere lengths, it is uncertain whether shortened telomeres are the exclusive cause of the phenotype. DC patients with DKC1 and TINF2 mutations usually present at a younger age than those with TERC or TERT mutations, and although they have more clinical abnormalities, there is no difference in telomere length between these patient subcategories [17]. Therefore, it is likely that pathways other than telomere maintenance are responsible for the disease phenotype.

For example, DKC1 and its associated NHP2, GAR1, NOP10 and CBF5 proteins interact with the H/ACA box of snoRNAs, which serves as a guide for pseudouridylation of ribosomal RNAs. Pseudouridylation is one of the hundred post-transcriptional modifications of RNAs (transfer, messenger, ribosomal and spliceosomal). As a result, defective dyskerin causes premature aging and acffects cell proliferation and haematopoietic potential and cancer [63]. In addition, other genes responsible for DC direct important functions other than those conected with telomere function. DKC1, TERT and TIN2 proteins are known to translocate to mitochondria, a crucial mechanism in modulating energy metabolism and ROS production under oxidative stress [64,65,66].

Due to the rarity of DC and the lack of living cells obtained from patients with specific mutations in diverse genes, researchers had to rely on animal models (zebrafish, Dictyostelium discoideum, mouse) to be able to study these mutations and dissect biochemical pathways and mechanisms involved in the disease. As increasingly demonstrated on these models, DKC1 depletion accelerates oxidative stress, which occurs prior to telomere shortening and affects ribosomal biogenesis which, in sequence, stimulates the p53 pathway [67].

## 5. Disease Progression

During follow-up, when the mucocutaneous triad is manifest, bone marrow failure is usually present. Sometimes, however, signs of the disease are vague and bone marrow failure or other abnormalities in another systems may present also before or in absence of the classical mucocutaneous traid [47,48,63]. Aplastic anemia, usually macrocytic with increased level of fetal hemoglobin, develops at an average age of onset of 11 years. It is associated with thrombocytopenia and then evolves to severe bone marrow failure [47,48,63]. Bone marrow failure can progress with the appearance of myelodysplasia in one or more clones [4]. Early death of can occur in 80% of cases of DC due to opportunistic infections. In DC, excessive shortening of telomeres is observed, which can lead to genome instability. Studies on electron microscopy have shown that certain DC cells have an immature embryonic core and that they can trigger a malignant transformation. In addition, the epithelial barrier is less effective than normal epithelium, so the permeability to harmful carcinogenic materials in the germ layer is higher. Leukoplastic areas can tipically undergo malignant transformation, and periodic monitoring is required. In addition, patients show a 40–50% cumulative risk incidence of malignancy at 50 years. DCS patients can develop Hodgkin’s lymphoma, larynx and bronchial cancer and GI tract adenocarcinoma, among others, of the genitourinary and skeletal system.

## 6. Differential Diagnosis

### 6.1. Disorders with Nail Dysplasia

Nail-patella syndrome (OMIM 161200)Twenty-nail dystrophy (OMIM 161050)Keratoderma with nail dystrophy and motor-sensory neuropathy (OMIM 148360)Poikiloderma with neutropenia (OMIM 604173)

### 6.2. Disorders with Reticulated Hyperpigmentation

The Naegeli-Franceschetti-Jadassohn syndrome (OMIM #161000) due to heterozygous pathogenic variants in the keratin-14 gene (*KRT14*) is distinguished, as it does not present leukoplakia, bone marrow disease, and an increased risk of malignancy. However, this disease, as well as reticular pigmentosum dermatopathy (OMIM #125595) [68], an allelic disorder, may show a similar reticulated hyperpigmentation.

Fanconi anemia (OMIM #227650) [69] usually shows diffuse or uniform pigment abnormalities, and pancytopenia appears earlier compared to DC [70]. Other common findings are eye and kidney disease and limb anomalies.

### 6.3. Disorders with Poikiloderma and Increased Photosensitivity

Other entities include Bloom syndrome (OMIM #210900), caused by mutations in *BLM* gene [71]; Rothmund-Thomson syndrome (OMIM #268400), most often due to variants in *RECQL4* gene [72]; and Epidermolysis Bullosa Simplex (OMIM #131900), due to mutations in 18 different genes [73], which present mottled pigmentation with similar poikiloderma. In addition, Kindler syndrome (OMIM #173650) due to *FERMT1* gene mutations [74], and poikiloderma with neutropenia type Clericuzio (OMIM #604173), associated with biallelic mutations in *USB1* gene [75]. In Bloom, Kindler, and Rothmund-Thomson syndromes, skin lesions may resemble those observed in DC, but they are more sensitive to the sun and have different related characteristics. Lastly, patients with graft-versus-host disease have poikiloderma, mucosal changes similar to lichen planus, and evident nail dystrophy after bone marrow transplantation [76].

### 6.4. Disorders with Bone Marrow Failure

Diamond-Blackfan anemia (DBA) is a disorder presenting with a profound isolated normochromic and usually macrocytic anemia with normal leukocytes and platelets; congenital malformations are observed in approximately 50% of affected individuals and growth retardation in 30%. Ninety percent of subjects with DBA present hematologic complications in the first year of life. DBA is associated with a higher risk of acute myelogenous leukemia, myelodysplastic syndrome and solid tumors. DBA is due to pathogenic variants in 16 genes encoding ribosomal proteins or in *GATA1* and *TSR2* genes. DBA generally shows an autosomal dominant inheritance; *GATA1*-related and *TSR2*-related DBA are inherited in an X-linked manner.

Shwachman-Diamond syndrome (SDS) is an AR disorder due to *SBDS* pathogenic variants and is characterized by exocrine pancreatic dysfunction with malabsorption, malnutrition, and growth failure, hematologic abnormalities with single- or multilineage cytopenia and susceptibility to myelodysplasia syndrome and acute myelogeneous leukemia, and bone abnormalities. Persistent or intermittent neutropenia is a common presenting finding, as well as short-stature and recurrent infections. Like DC, SDS may first present as bone marrow failure or gastro-intestinal malabsorption.

Differential diagnosis also includes other chromosomal breakage and reorganization syndromes such as Nijmegen break syndrome (OMIM #251260) [77], Seckel syndrome (OMIM #210600) [78], and finally the pseudo-TORCH syndrome (OMIM #251290) due to cerebral calcification [79].

## 7. Treatment

The integration of an interdisciplinary team is central to the management of DC and includes experts in dermatology, otorhinolaryngology, dentistry, maxillofacial surgery, oncology, gynecology and medical genetics, and including early genetic diagnostic facilities are crucial for timely family genetic counseling [4,59]. No definitive treatments are available for DC [13], and patients generally die prematurely of bone marrow failure [80]; allogeneic bone marrow transplantation is a treatment option, but it can be burdened by complications and risk of poor long-term survival. Up to 60% of DC patients with severe bone marrow failure may benefit temporarily from androgens or androgen-derivative therapy [81]. The biological mechanisms by which these compounds effectively treat bone marrow failure are not known. However, androgens can directly increase erythropoietin production or act on the erythropoietin receptor to elicit a hematologic response. Few studies on human cell lines and mouse models with aplastic anemia suggest that androgens can increase telomerase expression and, in turn, increase telomere length [82]. Furthermore, follow-up is crucial to detect the presence of tumors or severe infections due toopportunistic agents, which are among the main causes of death between the second and third decade of life [70].

The treatment of squamous cell carcinoma (SCC) of the head and neck is administered according to the anatomical region and stage of malignancy. Management of SCC can involve surgery, radiation and chemotherapy. In turn, exposure to potential carcinogens should be avoided, including ultraviolet radiation, alcohol and tobacco [83].

Systemic retinoids administered at low doses have determined some improvement in skin and nails in DC, but the side and long-term effects are uncertain [70,84].

Punch grafting is a low-cost and minimally invasive technique to enhance wound healing and has been associated with significant and quick pain reduction in DC ulcers [6,85].

Zoledronic acid treatment by intravenous injection was reported to prevent fractures in a young adult with DC [86]. According to an experimental study, zoledronic acid significantly increased bone volume and the number of hematopoietic stem cells in both young and adult mice [87].

Other exogenous therapies that can correct the telomerase defect and improve cell growth, as well as the use of modulators involved in telomere maintenance, have been suggested as new therapeutic methods for DC. Among them, the expression of a peptide derived from the dyskerin, a genetic suppressor element 24.2 (GSE24.2), which increases telomerase activity, regulates gene expression, and decreases DNA damage and oxidative stress in the cells of DC patients. Short peptides derived from GSE24.2 and GSE4, an eleven amino acid peptide have been shown to increase telomerase activity, reducing DNA damage as well as oxidative stress and cellular senescence in dyskerin-mutated cells. GSE4 expression also activated the c-myc and *TERT* promoters, as well as increased c-myc, *TERT* and *TERC* expression. Delivering GSE24.2, from a cDNA vector or as a peptide, reduces the pathogenic effects of *Dkc1* mutations in mice, thus suggesting a new therapeutic approach [88,89,90]. On the other hand, the therapeutic efficacy of telomerase activation using adeno-associated virus (AAV9) gene therapy vectors carrying the *Tert* gene for telomerase was tested in two independent mouse models of aplastic anemia due to short telomeres (*Trf1* and *Tert* deficient mice). A high dose of AAV9-*Tert* was found to target the bone marrow compartment, including hematopoietic stem cells. AAV9-*Tert* treatment after telomere attrition in bone marrow cells rescues aplastic anemia and mouse survival compared to mice treated with the empty vector. Improved survival is associated with a significant increase in telomere length in peripheral blood and bone marrow cells as well as better blood counts. These findings indicate that telomerase gene therapy represents a novel therapeutic strategy to treat aplastic anemia caused or associated with short telomeres [91].

## 8. Conclusions

DC is a clinically and genetically heterogeneous syndrome of bone marrow failure and a model of telomere disorders or telomeropathies. Different genetic alterations give rise to a wide spectrum of clinical symptoms with variable age of onset, so the diagnosis of DC can be a challenge in this disorder of telomere biology. Therefore, an interdisciplinary team is required. Available therapeutic options and timely family genetic counseling must be provided based on genetic diagnosis and telomere length laboratory studies.

## Figures and Tables

**Table 1 genes-13-00496-t001:** Genes responsible of abnormalities in telomere biology responsible of dyskeratosis congenita.

Mechanism of Action	Gene	Most Important Mutations	Inheritance Pattern
Telomerase holoenzyme complex	*DKC1* *	p.Ile38Thr, p.Thr49Met, p.Ser121Gly	XL
	*TERC* *	-	AD
	*TERT* *	p.Ala202Thr, p.His412Tyr	AD or AR
	*NOP10* *	p.Arg34Trp	AR
	*NHP2* *	p.Tyr139His, p.Val126Met, p.Ter154Arg	AR
Shelterin complex	*TPP1*	p.Lys170del, p.Pro491Thr	AD or AR
	*TINF2* *	p.Lys280Glu, p.Arg282His, p.Arg282Ser	AD
	*POT1*	p.Ser322Leu, p.Lys242Leu	AD
Telomere-limiting proteins	*CTC1*	p.Lys242Leufs * 41, p.Arg987Trp	AR
	*STN1*	p.Arg136Thr, p.Asp157Tyr	AR
Other proteins that interact directly or indirectly with key cellular processes	*RTEL1*	p.Arg981Trp	AD or AR
	*NAF1*	-	-
	*WRAP53*	p.Phe164Leu, p.Arg398Trp	AR
	*PARN*	p.Ala383Val, p.Asn288Lysfs * 23	AR

* Responsible for regulation of telomere length. AD: autosomal dominant. AR: autosomal recessive. XL: X-linked recessive.

## Data Availability

Refer to Michele Callea for any query, mcallea@gmail.com.

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
