# Peer review of "Multisystemic Manifestations in Rare Diseases: The Experience of Dyskeratosis Congenita"

_genes, 2022, doi:10.3390/genes13030496_

Round 1

Reviewer 1 Report

This is a well written and structured review of multisystemic manifestations of dyskeratosis congenita. This is an interesting review with concise data which adds to our current state of knowledge. However, there are several gaps that must be filled.

Major comments:

1. In Table 1 each of the epigraphs (telomerase holoenzyme complex, Telomere protection complex, etc) must be at the level of the first protein that begins to be enumerated.

TPP1 is a more common nomenclature than ACD and the authors should consider replacing it for a better understanding of the reader. Telomere protein complex should be replaced by shelterin complex (TINF2, POT1 and ACD = TPP1).

To complete the table, authors should include at least the most important mutations in each of the genes. For example, Ile38Thr, Thr49Met, Ser121Gly mutations in the DKC1 gene and so on.

2. Flow cytometry with fluorescent in situ hybridization is not the only method for the diagnosis of DC, although it is one of the most sensitive. The authors should comment that there are additional methods commonly used to measure telomere length, including Southern blot and RT-PCR to identify patients with DC.

3. DKC1 and TERT not only play a role in telomere maintenance but also have extratelomeric functions. The authors should also focus on these functions that could be involved in the development of the different forms of DC.

DKC1, together with its associated proteins NHP2, GAR1, NOP10 and CBF5, binds to the H/ACA box of snoRNAs, acting as a guide for the pseudouridylation of ribosomal RNAs. Pseudouridylation is one of the hundred post-transcriptional modifications of RNAs (transfer, messenger, ribosomal and spliceosomal). There are studies that have linked pseudouridylation with Alzheimer's or Parkinson's neurodegenerative diseases (Clin Biochem. 2007 Sep;40(13-14):936-8)

4. The authors should include that at the nuclear level, in addition to the telomere shortening and genomic instability, patients with DC present high DNA damage and oxidative stress.

5. The Hoyeraal-Hreidarsson syndrome is due to mutations in the RTEL1 and DKC1 genes as noted the authors but also to mutations in TINF2 (AD), TERT (AR), TPP1 (AR) and PARN (AR) genes.

6. Genes should be included in the Differential Diagnosis section as KRT14 in dermatopathia pigmentosa reticularis.

7. One of the main problems to investigate DC, is the living material that is available. Cells from patients with DC grow poorly and sometimes there are
mutations described in patients from whom cells could not be isolated. Therefore, researchers work with animal and cell models (zebrafish, Dictyostelium discoideum, mouse etc) that reproduce these mutations. These organisms evolutionarily distant from humans allow the study of biochemical pathways and mechanisms that are not deleterious in these models. The authors should emphasize this point and include a paragraph where it is discussed.

Additional comments:

  • Lines 48, 55, 56, 61, 99, 126, 195, 196, 201, 202: check spaces and commas.
  • Table 1: Change CD for DC. Include XL: X-linked reccesive in the footnote.
  • Lines 136, 203: reference.
  • Line 197: typo: pigmentation [Belligni 2011Fanconi anemia-
  • Line 217: delete underline-

Author Response

REVIEWER 1

This is a well written and structured review of multisystemic manifestations of dyskeratosis congenita. This is an interesting review with concise data which adds to our current state of knowledge. However, there are several gaps that must be filled.

Major comments:

1. In Table 1 each of the epigraphs (telomerase holoenzyme complex, Telomere protection complex, etc) must be at the level of the first protein that begins to be enumerated.

TPP1 is a more common nomenclature than ACD and the authors should consider replacing it for a better understanding of the reader. Telomere protein complex should be replaced by shelterin complex (TINF2, POT1 and ACD = TPP1).

To complete the table, authors should include at least the most important mutations in each of the genes. For example, Ile38Thr, Thr49Met, Ser121Gly mutations in the DKC1 gene and so on.

Thank you for the comment, the comment is amended and highlighted in the text

  1. Flow cytometry with fluorescent in situ hybridization is not the only method for the diagnosis of DC, although it is one of the most sensitive. The authors should comment that there are additional methods commonly used to measure telomere length, including Southern blot and RT-PCR to identify patients with DC.

Thank you for the comment, the commenti s amended and highlighted in the text

  1. DKC1 and TERT not only play a role in telomere maintenance but also have extratelomeric functions. The authors should also focus on these functions that could be involved in the development of the different forms of DC.

Thank you for the comment, the comment is amended and highlighted in the text

DKC1, together with its associated proteins NHP2, GAR1, NOP10 and CBF5, binds to the H/ACA box of snoRNAs, acting as a guide for the pseudouridylation of ribosomal RNAs. Pseudouridylation is one of the hundred post-transcriptional modifications of RNAs (transfer, messenger, ribosomal and spliceosomal). There are studies that have linked pseudouridylation with Alzheimer's or Parkinson's neurodegenerative diseases (Clin Biochem. 2007 Sep;40(13-14):936-8)

Thank you for the comment, the comment is amended and highlighted in the text

  1. The authors should include that at the nuclear level, in addition to the telomere shortening and genomic instability, patients with DC present high DNA damage and oxidative stress.

Thank you for the comment, the comment is amended and highlighted in the text

  1. The Hoyeraal-Hreidarsson syndrome is due to mutations in the RTEL1 and DKC1 genes as noted the authors but also to mutations in TINF2 (AD), TERT (AR), TPP1 (AR) and PARN (AR) genes.

Thank you for the comment, the comment is amended and highlighted in the text

  1. Genes should be included in the Differential Diagnosis section as KRT14 in dermatopathia pigmentosa reticularis.

Thank you for the comment, the comment is amended and highlighted in the text

  1. One of the main problems to investigate DC, is the living material that is available. Cells from patients with DC grow poorly and sometimes there are
    mutations described in patients from whom cells could not be isolated. Therefore, researchers work with animal and cell models (zebrafish, Dictyostelium discoideum, mouse etc) that reproduce these mutations. These organisms evolutionarily distant from humans allow the study of biochemical pathways and mechanisms that are not deleterious in these models. The authors should emphasize this point and include a paragraph where it is discussed.

Thank you for the comment, the comment is amended and highlighted in the text

Additional comments:

  • Lines 48, 55, 56, 61, 99, 126, 195, 196, 201, 202: check spaces and commas.
  • Table 1: Change CD for DC. Include XL: X-linked reccesive in the footnote.
  • Lines 136, 203: reference.
  • Line 197: typo: pigmentation [Belligni 2011Fanconi anemia-
  • Line 217: delete underline-

Reviewer 2 Report

  1. The word ‘Dyskeratosis’ in the manuscript title is misspelled as ‘Dyskeratoris’. Also, the title has two sentences separated by a period. Authors may want to consider using colon or dash instead of a period to make it as one sentence title.
  2. The manuscript has more than 30 simple punctuation mistakes that includes improper use of spacing, commas, period. The entire manuscript requires proofreading to correct these mistakes.
  3. Line 64: the name of gene is mentioned as ‘TERC8’. Number 8 does not exist in front of gene name TERC.
  4. Table 1: heading sentence for the table has ‘DC’ misspelled as ‘CD’.
  5. Table 1: The table has been grouped into three major groups of genes based on the mechanism of action. However, it is hard to figure out which genes belong which group due to absence of a line separating each group. For example, it is not clear from the table that genes NHP2 belong to ‘Telomerase holoenzyme complex’ or ‘Telomere protection complex’ group. Use either line or spacing between each group to make it easy for the readers to follow.
  6. In Clinical aspects section, authors have a subtitle named ‘XL: X-linked recessive’. It is not clear from the writeup that whether all three paragraphs that follow this title are associated with title or not.
  7. In Incidence section, authors report that ‘minus 1 in 1,000,000’. What is ‘minus’ here stand for? The cited paper (Dokal, 2011) does not have incidence number in that paper. Make sure include proper reference for this sentence. Additionally, only first sentence in ‘Incidence’ section is relevant to that section. Authors may want to consider expanding relevant information to this section or consider moving the current text to another section.
  8. Line 113: A period in the middle of a sentence.
  9. Change ‘X linked’ to X-linked’ at all places.
  10. Line 136: Citation missing
  11. Line 142: No period separating two sentences.
  12. Authors briefly mention about genotype-phenotype correlations in ‘Etiopathogenesis and diagnosis’ section. Author may want to consider expand this section little bit more.
  13. Section four (Evolution of DC). The word ‘Evolution’ may be misleading. Authors may want to consider another word to describe the same, such as ‘Disease progression’. References are not cited for many important points in this section. Authors may want to consider adding them.
  14. The ‘Clinical variants’ section could be moved up immediately after ‘Clinical aspects’ section. This would help the reader. Also, authors may want to consider using another word for ‘variants’. In genetics, variant word is commonly used these days to describe a particular gene mutation. Having the word ‘variant’ may confuse the readers.
  15. Line 178 and 179: It is a one sentence paragraph. Consider merging with previous paragraph or add at least one more sentence to make it as separate paragraph..
  16. Line 178: The genes RTEL1 and DKC1 are mentioned in this sentence. It appears like this is still related to Zinsser-Cole-Engman syndrome that they discuss in the previous paragraph. However, according to OMIM # 305000, Zinsser-Cole-Engman syndrome is caused by mutations in DKC1 gene only. But authors have listed both RTEL1 along with DKC1, why?
  17. Line 180: A period missing between two sentences
  18. Section ‘Differential diagnosis’: The words at the beginning of sentence (Line 193) appears to be from a middle of a sentence suggesting that earlier part is missing or deleted.
  19. Line 196: remove double commas
  20. Line 197: The reference ‘Belligni 2011’ is not listed in the Reference section. After listing, replace this reference with corresponding reference number.
  21. Section ‘Differential diagnosis’: The flow of information in this section is not that coherent. Authors may want to consider rewriting this section. They may to group differential diagnosis based on the major clinical features such as nail dysplasia, bone marrow failure syndromes, poikiloderma etc. Authors may want to consider adding Diamond-Blackfan anemia and Shwachman-Diamond syndrome to list of differentials.
  22. Line 222: ‘there is no cure of the disease’ is already mentioned earlier in the paragraph (line 216). No need to repeat the same sentence.
  23. Line 214: remove capital letter from word ‘Surgery’
  24. Line 236-238. Authors may want to consider expanding this paragraph as a paragraph should have minimum of two sentences.

Author Response

REVIEWER  2

  1. The word ‘Dyskeratosis’ in the manuscript title is misspelled as ‘Dyskeratoris’. Also, the title has two sentences separated by a period. Authors may want to consider using colon or dash instead of a period to make it as one sentence title.

Thank you for the comment, the comment is amended and highlighted in the text

  1. The manuscript has more than 30 simple punctuation mistakes that includes improper use of spacing, commas, period. The entire manuscript requires proofreading to correct these mistakes.

Thank you for the comment, the comment is amended and highlighted in the text

  1. Line 64: the name of gene is mentioned as ‘TERC8’. Number 8 does not exist in front of gene name TERC.
  2. Table 1: heading sentence for the table has ‘DC’ misspelled as ‘CD’.

Thank you for the comment, the comment is amended and highlighted in the text

  1. Table 1: The table has been grouped into three major groups of genes based on the mechanism of action. However, it is hard to figure out which genes belong which group due to absence of a line separating each group. For example, it is not clear from the table that genes NHP2 belong to ‘Telomerase holoenzyme complex’ or ‘Telomere protection complex’ group. Use either line or spacing between each group to make it easy for the readers to follow.

Thank you for the comment; the table seems quite straightforward to authors

  1. In Clinical aspects section, authors have a subtitle named ‘XL: X-linked recessive’. It is not clear from the writeup that whether all three paragraphs that follow this title are associated with title or not.
  2. In Incidence section, authors report that ‘minus 1 in 1,000,000’. What is ‘minus’ here stand for? The cited paper (Dokal, 2011) does not have incidence number in that paper. Make sure include proper reference for this sentence. Additionally, only first sentence in ‘Incidence’ section is relevant to that section. Authors may want to consider expanding relevant information to this section or consider moving the current text to another section.

Thank you for the comments, the comment is amended and highlighted in the text

  1. Line 113: A period in the middle of a sentence.

Thank you for the comment, the comment is amended and highlighted in the text

  1. Change ‘X linked’ to X-linked’ at all places.

Thank you for the comment, the comment is amended and highlighted in the text

  1. Line 136: Citation missing

Thank you for the comment, the comment is amended and highlighted in the text

  1. Line 142: No period separating two sentences.

Thank you for the comment, the comment is amended and highlighted in the text

  1. Authors briefly mention about genotype-phenotype correlations in ‘Etiopathogenesis and diagnosis’ section. Author may want to consider expand this section little bit more.

Thank you for the comment, the comment is amended and highlighted in the text

  1. Section four (Evolution of DC). The word ‘Evolution’ may be misleading. Authors may want to consider another word to describe the same, such as ‘Disease progression’. References are not cited for many important points in this section. Authors may want to consider adding them.

Thank you for the comment, the comment is amended and highlighted in the text

  1. The ‘Clinical variants’ section could be moved up immediately after ‘Clinical aspects’ section. This would help the reader. Also, authors may want to consider using another word for ‘variants’. In genetics, variant word is commonly used these days to describe a particular gene mutation. Having the word ‘variant’ may confuse the readers.

Thank you for the comment, the comment is amended and highlighted in the text

  1. Line 178 and 179: It is a one sentence paragraph. Consider merging with previous paragraph or add at least one more sentence to make it as separate paragraph..

Thank you for the comment, the comment is amended and highlighted in the text

  1. Line 178: The genes RTEL1 and DKC1 are mentioned in this sentence. It appears like this is still related to Zinsser-Cole-Engman syndrome that they discuss in the previous paragraph. However, according to OMIM # 305000, Zinsser-Cole-Engman syndrome is caused by mutations in DKC1 gene only. But authors have listed both RTEL1 along with DKC1, why?

Thank you for the comment, the comment is amended and highlighted in the text

  1. Line 180: A period missing between two sentences
  2. Section ‘Differential diagnosis’: The words at the beginning of sentence (Line 193) appears to be from a middle of a sentence suggesting that earlier part is missing or deleted.

Thank you for the comment, the comment is amended and highlighted in the text

  1. Line 196: remove double commas
  2. Line 197: The reference ‘Belligni 2011’ is not listed in the Reference section. After listing, replace this reference with corresponding reference number.
  3. Section ‘Differential diagnosis’: The flow of information in this section is not that coherent. Authors may want to consider rewriting this section. They may to group differential diagnosis based on the major clinical features such as nail dysplasia, bone marrow failure syndromes, poikiloderma etc. Authors may want to consider adding Diamond-Blackfan anemia and Shwachman-Diamond syndrome to list of differentials.

Thank you for the comment, the comment is amended and highlighted in the text

  1.  
  2. Line 222: ‘there is no cure of the disease’ is already mentioned earlier in the paragraph (line 216). No need to repeat the same sentence.

Thank you for the comment, the comment is amended and highlighted in the text

  1. Line 214: remove capital letter from word ‘Surgery’
  2. Line 236-238. Authors may want to consider expanding this paragraph as a paragraph should have minimum of two sentences.

Thank you

Round 2

Reviewer 1 Report

The review has gained strength and order after the revisions made by the authors.

The authors must include references that support the concepts of the following lines: 136, 149, 152, 155, 159, 166, 170, 235, 239 (Cells. 2019 Nov 8;8(11):1406.) (Blood. 2011 Nov 17;118(20):5458-65)(Genes Dev. 2008 Jul 1;22(13):1731-6.)

Lines 140 and 141 do not include references as numbers.

Author Response

Reviewer 2 comment

Comments and Suggestions for Authors

The review has gained strength and order after the revisions made by the authors.

The authors must include references that support the concepts of the following lines: 136, 149, 152, 155, 159, 166, 170, 235, 239 (Cells. 2019 Nov 8;8(11):1406.) (Blood. 2011 Nov 17;118(20):5458-65)(Genes Dev. 2008 Jul 1;22(13):1731-6.)

Done

Lines 140 and 141 do not include references as numbers.

We included reference number